# Targeting Asparagine Metabolism in Solid Tumors

**DOI:** 10.3390/nu17010179

**Published:** 2025-01-03

**Authors:** Keita Hanada, Kenji Kawada, Kazutaka Obama

**Affiliations:** 1Department of Gastrointestinal Surgery, Graduate School of Medicine, Kyoto University, Kyoto 606-8507, Japan; hanada@kuhp.kyoto-u.ac.jp (K.H.); kobama@kuhp.kyoto-u.ac.jp (K.O.); 2Department of Surgery, Rakuwakai Otowa Hospital, Kyoto 607-8062, Japan; 3Department of General Surgery, Kurashiki Central Hospital, Kurashiki 710-8602, Japan

**Keywords:** cancer, asparagine metabolism, glutamine metabolism, metabolic adaptation

## Abstract

Reprogramming of energy metabolism to support cellular growth is a “hallmark” of cancer, allowing cancer cells to balance the catabolic demands with the anabolic needs of producing the nucleotides, amino acids, and lipids necessary for tumor growth. Metabolic alterations, or “addiction”, are promising therapeutic targets and the focus of many drug discovery programs. Asparagine metabolism has gained much attention in recent years as a novel target for cancer therapy. Asparagine is widely used in the production of other nutrients and plays an important role in cancer development. Nutritional inhibition therapy targeting asparagine has been used as an anticancer strategy and has shown success in the treatment of leukemia. However, in solid tumors, asparagine restriction alone does not provide ideal therapeutic efficacy. Tumor cells initiate reprogramming processes in response to asparagine deprivation. This review provides a comprehensive overview of asparagine metabolism in cancers. We highlight the physiological role of asparagine and current advances in improving survival and overcoming therapeutic resistance.

## 1. Introduction

Asparagine is classified as a non-essential amino acid and plays a role in tumor progression. Asparagine depletion is an important therapeutic strategy. Indeed, L-asparaginase (ASNase) decreases plasma asparagine and is a key chemotherapeutic drug used against acute lymphoblastic leukemia (ALL). ASNase was first discovered in guinea pig serum as a component of dead lymphoma cells [1], and its active component is an enzyme which hydrolyzes asparagine to aspartate and ammonia [2,3]. When administered into the body, ASNase catalyzes asparagine in the blood and tissue fluids to aspartate, thereby depleting intracellular asparagine. It also has glutaminase activity, which degrades glutamine to glutamic acid and NH^4+^, partially depleting glutamine. On the other hand, asparagine is intracellularly synthesized from aspartate by asparagine synthase (ASNS) via a reaction that uses a single molecule of ATP and glutamine as a nitrogen source [4]. Most leukemia cells, such as ALL, lack ASNS activity and are dependent on exogenous asparagine for cell survival, which indicates that depletion of asparagine by ASNase leads to cell death [5,6]. Although ASNase is effective against ALL, solid tumors have been thought to be resistant to ASNase because of their ASNS activity [7]. Recently, therapies targeting ASNS activity and asparagine metabolism have been reported to be effective against some solid tumors in preclinical studies, and some clinical trials are ongoing. This review focuses on asparagine metabolism as a therapeutic target for solid tumors.

## 2. Activation of ASNS (Figure 1)

ASNS methylation has also been reported in solid tumors. Sugiyama and Andrulis et al. reported that the ASNS gene is silenced by DNA hypermethylation [8,9]. Peng et al. found that the complete methylation of CpG islands correlated with decreased mRNA expression in ASNase-sensitive 6C3HED cells (a strain requiring asparagine nutrition) [10]. Ren et al. reported that the induction of ASNS expression by ATF4 was not observed when the ASNS promoter was highly methylated [11]. Jiang et al. recently reported that promoter methylation of the *ASNS* gene prevents its transcriptional expression under asparagine depletion, which facilitates ATF4-independent induction of CCAAT-enhancer-binding protein homologous protein (CHOP) to trigger apoptosis [12]. Akagi et al. observed that methylation of CpG islands in the *ASNS* gene was frequently observed in T-cell ALL (T-ALL) [13]. Touzart et al. recently reported that promoter methylation of the *ASNS* gene was associated with ASNase sensitivity in both T-ALL cell lines and patient-derived xenografts, and that ASNS promoter methylation was a prognostic factor for patients with T-ALL [14]. Initially, ASNS protein expression was thought to determine ASNase susceptibility in ALL [6]; however, recent studies have shown that promoter methylation, along with low ASNS expression, may become a better biomarker for predicting ASNase sensitivity [14,15,16]. Analysis of metabolite profiling of 928 cell lines in the Cancer Cell Line Encyclopedia database indicated that lower asparagine levels elicited by the suppression of ASNS expression sensitize a subset of gastric cancer and hepatocellular carcinoma (HCC) cell lines, and that ASNS overmethylation results in a lack of ASNS protein expression, making them susceptible to ASNase both in vitro and in vivo [17]. Methylation of CpG islands regulates the basal expression level of ASNS. On the other hand, ASNS activity is also regulated by signaling pathways in response to cellular stresses: amino acid depletion and endoplasmic reticulum stress [18]. The response pathways to amino acids are activated in response to intracellular amino acid depletion or imbalance in amino acid homeostasis [19,20].

Amino acids are essential for rapidly proliferating cancer cells because they are required for the synthesis of lipids, nucleic acids, and proteins [21]. General control nonderepressible 2 (GCN2) is a serine/threonine protein kinase which senses amino acid deprivation by binding to uncharged transfer RNAs, thereby serving as a main regulator of the amino acid response [22,23,24]. Endoplasmic reticulum stress increases ASNS transcription via activation of PKR-like endoplasmic reticulum-resident kinase (PERK) in this stress response pathway [25]. Both GCN2 kinase and PERK activation cause phosphorylation of eukaryotic initiation factor 2α (eIF2α) and increase levels of the transcription factor activating transcription factor 4 (ATF4), which induces ASNS transcription [26,27,28]. Hinze et al. reported that sensitization to ASNase is mediated by Wnt-dependent stabilization of proteins, which inhibits glycogen synthase kinase 3-dependent protein ubiquitination and proteasomal degradation in leukemia cells [29].

Glutamine is the most abundant and widely used amino acid in the body and is an indispensable energy substrate for maintaining tumor progression, and its downstream metabolite, asparagine, is also important for tumors [30]. The relationship between asparagine and glutamine metabolism has been reported. ASNase-resistant human leukemia cells exhibit elevated glutamine synthetase activity [31]. Tardito et al. reported that in human sarcoma cells, glutamine synthetase contributed to the adaptation of tumor cells to ASNase, and that inhibition of glutamine synthetase enhanced the effect of ASNase [32]. Chiu et al. reported that combination therapy with ASNase and glutamine synthetase inhibitors completely inhibited glutamine metabolism and enhanced antitumor effects in HCC [33]. Glutamine is the most abundant amino acid in human serum, and many cancer cells consume large amounts of glutamine to meet their energy needs, which contributes to glutamine depletion [34,35,36]. By contrast, Zhang et al. found that apoptosis induced by glutamine starvation was rescued only by asparagine supplementation among various amino acids, indicating that asparagine plays an important role in regulating cellular adaptation to glutamine deficiency [37].

## 3. Asparagine Metabolism in Solid Tumors

The significance of ASNS expression in solid tumor progression and its sensitivity to ASNase has not yet been well studied, but has gradually received attention since the 2000s. The following text describes asparagine metabolism for each solid tumor.

### 3.1. Ovarian Cancer

Scherf et al. analyzed a cDNA microarray from 60 human cancer cell lines and reported a negative correlation between mRNA levels of ASNS and sensitivity to ASNase treatment in ovarian cancer cell lines as well as ALL cell lines [38]. Lorenzi et al. also reported that there was a negative correlation between mRNA levels of ASNS and sensitivity to ASNase in ovarian cancer cells, and that the knockdown of ASNS expression with siRNA methods increased the susceptibility to ASNase [39]. Subsequent studies using a number of ovarian cancer cell lines also reported a negative correlation between protein levels of ASNS and ASNase efficacy [40].

### 3.2. Pancreatic Ductal Cell Adenocarcinoma (PDCA)

ASNS expression is highest in the pancreas among any organs, and within the pancreas, ASNS is mainly present in the acinar cells [41]. Dufour et al. reported that while normal pancreatic exocrine cells express high levels of ASNS protein, screening of 98 human PDCA cells revealed that ASNS protein was not detected in about 70% of them, suggesting the possibility of ASNase therapy for PDCA [42]. Both in vitro and in vivo experiments indicated that PDCA cell lines with lower expression of ASNS were more sensitive to ASNase. Cui et al. reported that glucose deprivation increased ASNS expression and was associated with enhanced resistance to cisplatin-induced apoptosis in PDCA cells [43]. Regarding translational reprogramming of asparagine-restricted cancer cells, Pathria et al. reported that asparagine restriction in PDCA cells activated mitogen-activated protein kinase (MAPK) signaling, mammalian target of rapamycin complex 1 (mTORC1), and eukaryotic initiation factor 4E (eIF4E), which resulted in enhanced translation of ATF4. MAPK inhibition attenuates the translational induction of ATF4 and its target ASNS, sensitizing pancreatic tumors to asparagine restriction and thus inhibiting tumor growth. These studies provide a rationale for combination therapy with MAPK inhibitors and ASNase against PDCA cells [44]. Using cell panel analysis, Nakamura et al. found that several PDCA cell lines were highly sensitive to combination therapy with GCN2 inhibitors and ASNase [45].

### 3.3. Gastric Cancer

Yu et al. reported that ASNS knockdown in gastric cancer cell lines conferred suppressed tumor growth and sensitivity to cisplatin, and that low ASNS expression was significantly associated with improved survival in patients with gastric cancer [46]. These data reveal that ASNS can become a novel therapeutic target for gastric cancer. Wang et al. reported that Tousled-like kinase 2 (TLK2) is correlated with the mTORC1/ASNS axis and regulates metabolism in gastric cancer, which makes TLK2 a potential therapeutic target [47]. Recently, Li et al. reported that transmembrane protein 176B (TMEM176B) regulates ASNS via the phosphatidylinositol 3-carboxykinase (PI3K)–protein kinase B (AKT)–mTOR signaling axis in gastric cancer [48].

### 3.4. Lung Cancer

Xu et al. reported that ASNS gene expression was higher in lung cancer tissues than in normal tissues, and that ASNS knockdown in human lung cancer cell lines inhibited cell growth by arresting the cell cycle in the G0/G1 phase [49]. Gwinn et al. reported that, in non-small-cell lung cancer (NSCLC), the PI3K-AKT pathway, downstream of KRAS signaling, modulates the activity of NF-E2-related factor 2 (NRF2) to regulate ATF4 and the ATF4 target ASNS, which results in asparagine biosynthesis and amino acid uptake during nutrient depletion. In addition, combination therapy involving the depletion of extracellular asparagine (e.g., ASNase) and AKT inhibition decreased tumor growth, suggesting that ASNS can be a promising therapeutic target for KRAS-driven NSCLC [50]. Cai et al. reported that ASNS promoted metastasis in the absence of endogenous asparagine and was dependent on the Wnt pathway and mitochondrial function in lung cancer cell lines [51]. Deng et al. reported that Aurora kinase A (AURKA) regulated amino acid synthesis and was a vulnerable target in Kelch-like ECH-associated protein 1 (KEAP1)-deficient NSCLC [52]. Mechanistically, AURKA interacts with eIF2α kinase GCN2 and maintains its phosphorylation to regulate amino acid biosynthesis via eIF2α-ATF4, whose inhibition suppresses ASNS expression.

### 3.5. Breast Cancer

Pacher et al. reported that insulin-like growth factor is highly expressed in breast cancer, and that ASNS is one of the targets of insulin-like growth factor signaling [53]. Yang et al. reported that ASNS knockdown inhibited cell proliferation in breast cancer cells by arresting the cell cycle in the S phase [54]. Pavlova et al. reported that asparaginase blocks the ability of several human breast cancer cells to adapt to glutamine deprivation, indicating that asparagine can become an essential amino acid when extracellular glutamine declines [55]. Furthermore, Knott et al. reported that ASNS expression and asparagine utilization were strongly associated with the metastatic behavior of breast cancer [56]. Under amino acid-deficient culture conditions, asparagine supplementation greatly enhanced the invasive potential of breast cancer cells; in vivo, ASNS knockdown and ASNase or dietary asparagine restriction did not change the tumor growth of primary tumors but significantly reduced the metastatic lung nodules. Using circulating tumor cells from mice harboring a xenograft of triple-negative breast cancer cells, Ameri et al. found that, under hypoxic conditions, circulating tumor cells increased the expression of ATF3, ATF4, and ASNS and exhibited a more aggressive phenotype [57]. When analyzing published datasets, Lin et al. reported that triple-negative breast cancer exhibited the highest expression levels of ASNS protein among breast cancer subtypes [58].

### 3.6. Colorectal Cancer (CRC)

Ye et al. reported that ATF4 knockdown decreased CRC cell survival and proliferation, and that overexpression of ASNS or asparagine supplementation reversed the growth inhibition and increased survival of ATF4-knockdown cells, which suggests that the GCN2-eIF2α-ATF4 pathway is important for metabolic homeostasis [59]. Lin et al. evaluated ASNS expression in the tumor tissues of 172 rectal cancer patients using immunohistochemical analysis and found that ASNS deficiency was a poor prognostic predictor for metastasis-free survival and disease-specific survival. Furthermore, ASNS deficiency was associated with reduced treatment response and worse survival in rectal cancer patients treated with neoadjuvant chemoradiotherapy [60].

Oncogenic mutations in the *KRAS* gene are found in about 40% of CRC cases. Toda et al. reported that intracellular asparagine synthesis by ASNS is an important mechanism of resistance to glutamine deficiency in KRAS-mutant CRC cell lines, and that ASNS protein expression is higher in tumor tissues from patients with KRAS-mutant CRC than in those from patients with KRAS wild-type CRC. They also reported that ASNS expression is induced by KRAS activation signaling, particularly via PI3K-AKT-mTOR signaling, and that a combination of ASNase and mTOR inhibitor suppresses the growth of KRAS-mutant CRC cells in vivo [61]. Furthermore, using a patient-derived spheroid xenograft, Nishikawa et al. found that delivering ASNase treatment to ASNS-knockdown spheroids exhibited dramatically suppressed tumor engraftment, suggesting that ASNS inhibition can be promising in targeting asparagine metabolism [62].

Macropinocytosis is a receptor-independent, non-selective endocytic pathway used for nutrient acquisition from extracellular proteins, lipids, and cell debris. In KRAS-transformed cancer cells, enhanced macropinocytosis is recognized under nutrient stress conditions [63]. Hanada et al. reported that the combination of ASNS knockout and ASNase inhibits the growth of KRAS-mutant CRC cells, and that the combination therapy of asparagine depletion and macropinocytosis inhibition dramatically suppresses tumor growth of KRAS-mutant CRC cells [64]. Du et al. analyzed the expression of sex-determining region Y (SRY)-box transcription factor 12 (SOX12) in a CRC cohort and found that overexpression of SOX12 was correlated with poor prognosis and metastatic behavior in female CRC patients [65]. In CRC cell lines, SOX12 promoted asparagine synthesis by activating glutaminase, glutamate transaminase 2 (GOT2), and ASNS. They further reported that SOX12 expression was positively correlated with the expression of glutaminase, GOT2, ASNS, and hypoxia-inducible factor-1α in human CRC clinical samples. Aladelokun et al. found that inhibition of ASNS signaling synergistically inhibits CRC progression with G protein-coupled estrogen receptor, which may result in poor survival for female CRC patients with high ASNS [66]. Recently, clinical studies focusing on sex differences have shown that asparagine metabolism is enhanced in CRC, especially in women, and increased ASNS expression is correlated with decreased survival in women [67,68]. Deng et al. reported that p53 regulates aspartate–asparagine homeostasis via transcriptional expression of ASNS, resulting in inhibition of CRC cell growth in vitro and in vivo. Mechanically, aspartate–asparagine homeostasis regulates AMPK-mediated p53 activation by modulating LKB1 activity [69]. Hinze et al. reported that GSK3α inhibition was sufficient for ASNase sensitization in APC or β-catenin mutant CRC, although ASNase had little effect on APC or β-catenin mutant CRC [70]. Regarding the metabolic differences between adenoma and adenocarcinoma, Legge et al. reported that adenoma cells were largely resistant to asparagine depletion, while late-stage adenocarcinoma cells were dependent on ASNS to support mTORC1 signaling and maximal glycolytic and oxidative capacity [71].

### 3.7. Prostate Cancer

Patrikainen et al. reported that ASNS is one of the overexpressed genes in prostate cancer cell lines adapted to grow in a floating state [72]. Overexpression of ASNS mRNA was detected in integrated analysis of surgically resected specimens from castration-resistant prostate cancer and was associated with ASNS protein levels [73]. Moreover, ASNS protein expression was correlated with treatment resistance. Interestingly, ASNase action and ASNS induction in prostate cancer cell lines were used to validate a detection system that measured amino acid restriction in tumors based on ribosome profiling [74]. Using transcriptomic and metabolomic analyses, Yoo et al. found that prostate cancer with altered TP53 had an increased dependence on asparagine and overexpression of ASNS [75].

### 3.8. Hepatocellular Carcinoma (HCC)

Somewhat conflicting results have been reported regarding the role of ASNS in human HCC. Zhang et al. reported that ASNS is overexpressed in HCC and is associated with serum α-fetoprotein levels, tumor size, stage, and vascular invasion; however, patients with lower ASNS expression exhibited poorer overall survival outcomes. In experiments with HCC cell lines, ASNS inhibited tumor cell proliferation, migration, and tumorigenesis [76]. Li and Dong reported that ASNS levels, along with the levels of the endoplasmic reticulum stress-related transcription factor ATF6, were lower in HCC patients than in control patients or chronic hepatitis B patients [77]. Using a mouse model of HCC, Zhou et al. recently reported that pituitary tumor transforming gene 1 [PTTG1] deficiency significantly suppressed the development of HCC. The mechanism is that PTTG1 binds to the promoter of ASNS, which promotes ASNS transcription, increases asparagine levels, activates the mTOR pathway, and promotes HCC progression [78]. Mossmann et al. also reported that arginine levels were elevated in HCC, and that ASNS upregulation via RNA binding motif protein 39 (RBM39) promoted arginine uptake [79].

### 3.9. Other Malignancies

Recently, there have been reports on the signaling pathways involved in regulating ASNS expression and the antitumor effects of ASNS inhibition. Using glioblastoma cells, Zhang et al. reported that glutamine is a critical amino acid for cell survival, and that glutamine withdrawal-induced apoptosis was rescued by asparagine among various amino acids, indicating that ASNS is required for glutamine-dependent survival [37]. Li et al. reported that ASNS knockdown suppressed cell proliferation in human epidermoid carcinoma cells and melanoma cells [80]. Krall et al. reported that asparagine is used as an exchange factor between intracellular asparagine and extracellular amino acid uptake to promote cancer cell growth in Hela and epidermoid carcinoma cells. They found that only intracellular asparagine is exchanged with extracellular amino acids, particularly serine, arginine, and histidine, to promote protein and nucleotide synthesis [81]. Ye et al. reported that ATF4 knockdown decreased the survival and proliferation of sarcoma cell lines, and that decreased proliferation and increased apoptosis were associated with decreased ASNS expression in ATF4-deficient cells. Importantly, asparagine supplementation for ATF4-knockdown cells rescued cell survival [59]. In osteosarcoma, Zheng et al. reported that nuclear ubiquitous casein and cyclin-dependent kinase substrate 1 (NUCKS1) promoted asparagine synthesis by transcriptionally upregulating ASNS expression, which promoted cell growth and metastasis [82].

The signal pathways and processes related to ASNS and asparagine metabolism are listed in Table 1. Therapeutic reports targeting ASNS against solid tumors are listed in Table 2.

## 4. ASNS, Autophagy and Macropinocytosis (Figure 2)

Autophagy consists of the intracellular lysosomal degradation and recycling of proteins and organelles and is essential to maintain metabolic energy homeostasis and survival under starvation. Several reports have been published on the link between ASNS and autophagy; Takahashi et al. reported that ASNase treatment led to metabolic arrest via reductions in both oxidative phosphorylation and glycolysis, along with mitochondrial damage and autophagy. Autophagy attempts to rescue cells by reducing reactive oxygen species (ROS) levels through the elimination of damaged mitochondria [83]. Lin et al. reported that asparagine deprivation and ASNS expression were enhanced by the conditional knockout of ATG5 in a mouse model of salivary duct carcinoma driven by oncogenic KRASG12V [58]. In ALL, glioblastoma, ovarian cancer, and CRC cells, inhibition of the autophagy pathway by chloroquine administration in combination with ASNase has been reported to enhance cell growth inhibition [83,84,85,86]. Some aspects of autophagy are functionally similar to macropinocytosis, a vesicular pathway that mediates the degradation of extracellular albumin. Hanada et al. reported that ASNS inhibition induces macropinocytosis, leading to extracellular albumin uptake and amino acid degradation. They also observed that, in addition to asparagine depletion, combination therapy with macropinocytosis inhibition dramatically inhibits tumor growth of KRAS-mutant CRC cells in vivo [64].

## 5. ASNS Inhibitors and ASNase

Ikeuchi et al. reported that amino sulfoximine 5 (AS5) acts as an ASNS inhibitor, and that AS5 alone or in combination with ASNase inhibited cell proliferation in ALL cells [87]. Hettmer et al. also reported that AS5 inhibited in vivo and in vitro cell proliferation in mouse and human sarcoma cell lines [88]. They also found that ASNS knockdown in mouse and human sarcoma cell lines decreased the percentage of S-phase cells, which was rescued by the addition of exogenous asparagine. Recently, Pan et al. reported that bisabosqual A was discovered as a novel ASNS inhibitor from applying in vitro screening, and that the combination of bisabosqual A and ASNase suppressed cell proliferation of a human NSCLC cell line [89]. Zhu et al. reported that the high-resolution crystal structure of ASNS can be used to identify a domain of inhibitor binding and selectivity, which can result in the discovery of second-generation ASNS inhibitors [90].

The combination of ASNase and chemotherapy has been reported to improve progression-free survival and overall survival in advanced pancreatic adenocarcinoma, regardless of ASNS expression levels [91]. Phase III trials based on these results are currently ongoing (Table 3).

## 6. Conclusions

This article outlines the role of asparagine in tumor progression and explores the impact of asparagine bioavailability in targeted tumor therapy. Targeting the signaling pathways involved in asparagine metabolism allows for precise therapeutic approaches to several types of cancers.

## Figures and Tables

**Figure 1 nutrients-17-00179-f001:**
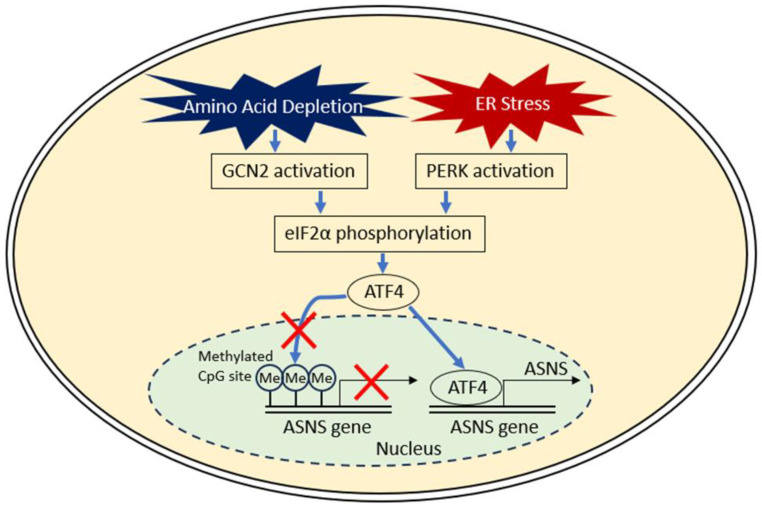
Regulatory mechanism of ASNS expression. ASNS activity is regulated by signaling pathway in response to cellular stresses: amino acid depletion and endoplasmic reticulum (ER) stress. GCN2, general control nonderepressible 2; eIF2α, eukaryotic initiation factor 2α; ATF4, activating transcription factor 4; PERK, PKR-like endoplasmic reticulum-resident kinase.

**Figure 2 nutrients-17-00179-f002:**
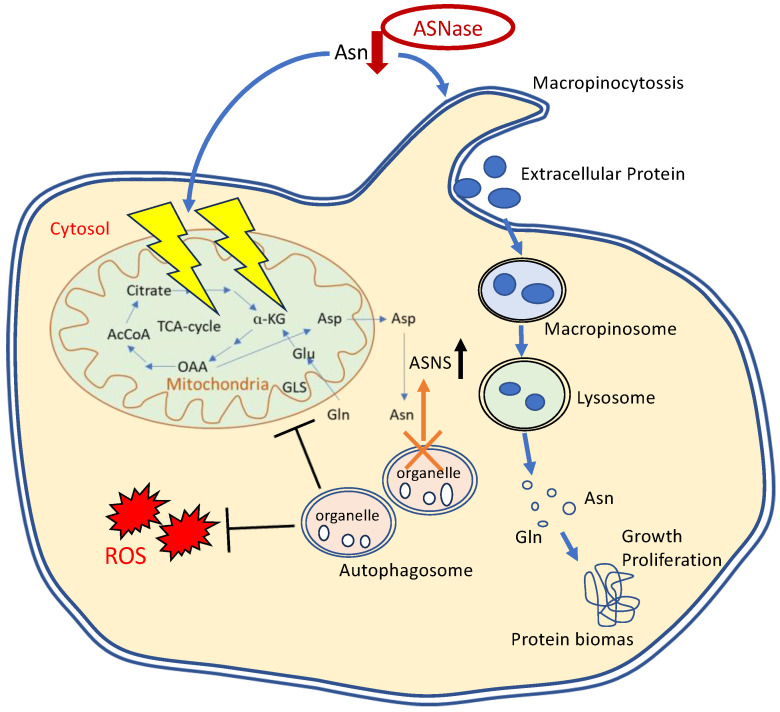
Regulation of intracellular asparagine levels by ASNS, autophagy, and macropinocytosis. Cancer cells exposed to ASNase prevent cell death by inducing autophagy and macropinocytosis. Autophagy attempts to rescue cancer cells by reducing reactive oxygen species (ROS) levels through the elimination of damaged mitochondria. Cancer cells respond to asparagine depletion by inducing macropinocytosis, uptake of extracellular albumin, and amino acid degradation.

**Table 1 nutrients-17-00179-t001:** Signaling pathways and processes related to ASNS and asparagine metabolism in solid tumors.

Cancer Type	Signaling Pathways and Processes Related to ASNS and Asparagine Metabolism	Reference
Pancreatic Ductal Cell Adenocarcinoma (PDCA)	MAPK, mTORC1, eIF4E, ATF4	[44]
	GCN2	[45]
Gastric cancer	TLK2	[47]
	TMEM176B, PI3K, Akt, mTOR	[48]
Lung cancer	PI3K, AKT, NRF2, ATF4	[50]
	Wnt	[51]
	AURKA, GCN2, eIF2α, ATF4	[52]
Breast cancer	ATF3, ATF4	[57]
	EMT	[56]
Colorectal cancer (CRC)	GCN2, eIF2α, ATF4	[59]
	KRAS, PI3K, AKT, mTOR	[61]
	Macropinocytosis	[64]
	SOX12	[65]
	p53	[69]
	GSK3α	[70]
	mTORC1	[71]
Prostate Cancer	TP53	[75]
Hepatocellular Carcinoma (HCC)	ATF6	[77]
	PTTG1, mTOR	[78]
	RBM39	[79]
Fibrosarcoma	GCN2, eIF2α, ATF4	[59]
Osteosarcoma	NUCKS1	[82]

**Table 2 nutrients-17-00179-t002:** Therapeutic reports targeting ASNS against solid tumors.

Year	Author	Cancer Type	ASNS Inhibition	Reference
2010	Ye	Sarcoma	shATF4, shGCN2	[59]
2016	Toda	Colorectal cancer (CRC)	shASNS, mTOR inhibitor + ASNase	[61]
2018	Knot	Breast cancer	shASNS + ASNase	[56]
2018	Gwinn	Non-small-cell lung cancer (NSCLC)	ASNS knockout, AKT inhibitor + ASNase	[50]
2018	Nakamura	Pancreatic cancer	GCN2 inhibitor + ASNase	[45]
2019	Pathria	Pancreatic cancer, Melanoma	MEK inhibitor + ASNase	[44]
2021	Hanada	Colorectal cancer (CRC)	ASNS knockout + ASNase	[64]
2022	Nishikawa	Colorectal cancer (CRC)	ASNS knockout + ASNase	[62]

**Table 3 nutrients-17-00179-t003:** Clinical trials on L-Asparaginase in solid tumors.

Trial Start (Year)	Cancer Type	Phase	Trial Number	Regimen	Primary Outcomes	Results
2014	Pancreatic cancer	II	NCT02195180	Chemotherapy + ASNase (ERY001) vs. Chemotherapy alone	OS *, PFS **	Improvements in OS * and PFS **
2018	Pancreatic cancer	III	NCT03665441	Chemotherapy + ASNase (Eryaspase) vs. Chemotherapy alone	OS *	Ongoing
2019	Breast cancer	II/III	NCT03674242	Chemotherapy + ASNase (Eryaspase) vs. Chemotherapy alone	DCR ***	Ongoing
2022	Solid tumor	I	NCT05631327	single-agent ASNase (JZP341)	DCR *** (Dose Expansion Phase)	Ongoing

* OS, overall survival; ** PFS, progression-free survival; *** DCR, disease control rate.

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
