# Peer review of "Targeting Asparagine Metabolism in Solid Tumors"

_nutrients, 2025, doi:10.3390/nu17010179_

Round 1
Reviewer 1 Report
Comments and Suggestions for Authors
1) This review article aims to provide an overview of asparagine metabolism and its physiological role in cancer, as well as the current advancements in improving survival and overcoming therapeutic resistance. However, the information is not presented clearly. It is recommended to include illustrations explaining the contents in each paragraph to enhance readers' understanding.
- Particularly, Section 2. Activation of ASNS, Section 4. Asparagine Metabolism in Solid Tumors, and Section 5. ASNS and Autophagy
2) In section 2,
Rather than vaguely mentioning that ASNS is highly expressed in each cancer type, it seems necessary to specifically explain, why targeting ASNS in each cell suggests therapeutic potential.
3) Check some errors.
- Please use abbreviation after defining (line 64, HCC; line 193, AFP).
- Line 81: Check the sentence (period is located in wrong location).
- Lines 27: It is written, “depletion of asparagine by ASNS”. Isn’t it ASNase, not ASNS?
4) Should Table 1 be located under the title of Section 3. Asparagine Metabolism in Solid Tumors?
If it should be, please correct the numbering of references in Table 1 so that they are in order.
5) The titles in Section 3 and Section 4 are same, “Asparagine Metabolism in Solid Tumors”. Please correct them.
Author Response
A point-by-point response to the reviewers (nutrients-3370793)
Reviewers’ comments are quoted “verbatim” >> followed by our response.
“Reviewer #1
1) This review article aims to provide an overview of asparagine metabolism and its physiological role in cancer, as well as the current advancements in improving survival and overcoming therapeutic resistance. However, the information is not presented clearly. It is recommended to include illustrations explaining the contents in each paragraph to enhance readers' understanding.
- Particularly, Section 2. Activation of ASNS, Section 4. Asparagine Metabolism in Solid Tumors, and Section 5. ASNS and Autophagy”
>> Thank you for your insightful comments. As requested, we have revised the text as well as added two figures (Figure 1, and Figure 2) and one table (Table 1).
“2) In section 2,
Rather than vaguely mentioning that ASNS is highly expressed in each cancer type, it seems necessary to specifically explain, why targeting ASNS in each cell suggests therapeutic potential.”
>> Thank you for your insightful comments. I think your point is valid. In order to facilitate readers’ understanding, we have revised the text by adding the content of Section 4 of the original manuscript to Section 2. The revised sections are in yellow background text.
“3) Check some errors.
- Please use abbreviation after defining (line 64, HCC; line 193, AFP).
- Line 81: Check the sentence (period is located in wrong location).
- Lines 27: It is written, “depletion of asparagine by ASNS”. Isn’t it ASNase, not ASNS?” 
>> We apologize for the inadequate description. We have corrected the text.
“4) Should Table 1 be located under the title of Section 3. Asparagine Metabolism in Solid Tumors?
If it should be, please correct the numbering of references in Table 1 so that they are in order.”
>> In the revised text, we have added one Table (Table 1), and slightly changed the position of the Tables (Table 1 and 2).
“5) The titles in Section 3 and Section 4 are same, “Asparagine Metabolism in Solid Tumors”. Please correct them. “
>> We apologize for the inadequate description. Section 4 was actually titled “ASNS Methylation in Solid Tumors”. However, due to content, the text of Section 4 was merged into Section 2 in the revised text.

Reviewer 2 Report
Comments and Suggestions for Authors
In this manuscript, the authors reviewed asparagine metabolism-targeting strategies for overcoming cancer therapeutic resistance. This is a comprehensive review, and I believe it will be useful for those working in related fields. However, minor revisions are still needed before acceptance.
- It would benefit readers if the authors included more detailed information in the diagrams/schemes.
- Parts 3 and 4 have the same subtitle.
- Parts 5 and 6 could be integrated into Parts 3 and 4.
- The logic of this review could be optimized. The authors could describe how ASNS blocking enhances other treatments, such as chemotherapy, radiation, or immunotherapy.
Author Response
A point-by-point response to the reviewers (nutrients-3370793)
Reviewers’ comments are quoted “verbatim” >> followed by our response.
“Reviewer #2
In this manuscript, the authors reviewed asparagine metabolism-targeting strategies for overcoming cancer therapeutic resistance. This is a comprehensive review, and I believe it will be useful for those working in related fields. However, minor revisions are still needed before acceptance.
- It would benefit readers if the authors included more detailed information in the diagrams/schemes.”
>> Thank you for your positive comment. As requested, we have added two figures (Figure 1, and Figure 2) and one Table (Table 1) to facilitate readers’ understanding.
“2. Parts 3 and 4 have the same subtitle.” 
>> We apologize for the inadequate description. Section 4 was actually titled “ASNS Methylation in Solid Tumors”. However, due to content, the text of Section 4 was merged into Section 2 in the revised text.
“3. Parts 5 and 6 could be integrated into Parts 3 and 4.”
>> Thank you for your comment. However, we believe that ASNS, Autophagy, and ASNS inhibitors should be separated in terms of content.
“4. The logic of this review could be optimized. The authors could describe how ASNS blocking enhances other treatments, such as chemotherapy, radiation, or immunotherapy.”
>> We thank Reviewer #2’s comment. As far as we know, ASNS inhibition is still a work in progress, and there seems to be no clear report on the enhancement of chemotherapy, radiotherapy, and immunotherapy. I think that is an issue for the future.

Reviewer 3 Report
Comments and Suggestions for Authors
I am pleased to provide a review of this manuscript.
The review provides an insightful and comprehensive overview of asparagine metabolism in cancer, effectively summarizing its physiological roles, involvement in tumor progression, and recent advancements in improving survival and overcoming therapeutic resistance. The discussion on the impact of asparagine bioavailability on targeted tumor therapy and the potential of targeting asparagine-related signaling pathways as precise therapeutic strategies is particularly well-articulated and valuable.
I have no major criticisms.
Comments on the Quality of English LanguageWhile the content is thorough and informative, there are numerous minor grammatical issues throughout the manuscript that need to be addressed to enhance clarity and readability. The examples provided in this review are not exhaustive but serve to highlight the types of errors present. We strongly recommend that the authors carefully review the entire manuscript to identify and correct all grammatical issues beyond those listed here.
Examples of grammar issues in Abstract part:
Line 11-12: A comma is needed after "addiction" to separate the appositive phrase correctly.
Line 15-16: The phrase "is often as an anticancer strategy" is awkward and grammatically incorrect. Remove "as" for clarity.
Line 18-19:"Discuss a comprehensive overview" is redundant. Instead, use either "provide a comprehensive overview" or "specifically discuss."
Author Response
A point-by-point response to the reviewers (nutrients-3370793)
Reviewers’ comments are quoted “verbatim” >> followed by our response.
“Reviewer #3
Comments and Suggestions for Authors
I am pleased to provide a review of this manuscript.
The review provides an insightful and comprehensive overview of asparagine metabolism in cancer, effectively summarizing its physiological roles, involvement in tumor progression, and recent advancements in improving survival and overcoming therapeutic resistance. The discussion on the impact of asparagine bioavailability on targeted tumor therapy and the potential of targeting asparagine-related signaling pathways as precise therapeutic strategies is particularly well-articulated and valuable.
I have no major criticisms.”
>> We appreciate the reviewer’s insightful comment.
“Comments on the Quality of English Language
While the content is thorough and informative, there are numerous minor grammatical issues throughout the manuscript that need to be addressed to enhance clarity and readability. The examples provided in this review are not exhaustive but serve to highlight the types of errors present. We strongly recommend that the authors carefully review the entire manuscript to identify and correct all grammatical issues beyond those listed here.
Examples of grammar issues in Abstract part:
Line 11-12: A comma is needed after "addiction" to separate the appositive phrase correctly.
Line 15-16: The phrase "is often as an anticancer strategy" is awkward and grammatically incorrect. Remove "as" for clarity.
Line 18-19:"Discuss a comprehensive overview" is redundant. Instead, use either "provide a comprehensive overview" or "specifically discuss."
>> We apologize for the inadequate description. As requested, we have corrected the text.

Round 2
Reviewer 1 Report
Comments and Suggestions for Authors
Figures need legends, not with just simple titles. Figure legend should include definitions of abbreviations used and a brief summary of the Figure.
Figures 1 and 2 also need to be modified. It is preliminary.
Author Response
A point-by-point response to the reviewers (nutrients-3370793)
Reviewers’ comments are quoted “verbatim” >> followed by our response.
“Comments and Suggestions for Authors
Figures need legends, not with just simple titles. Figure legend should include definitions of abbreviations used and a brief summary of the Figure.
Figures 1 and 2 also need to be modified. It is preliminary.”
>> Thank you for your comment. As requested, we have added Figure legends (for Figure 1 and Figure 2) and modified two figures (Figure 1 and 2).
